# Clinical Characterization of Targetable Mutations (BRAF V600E and KRAS G12C) in Advanced Colorectal Cancer—A Nation-Wide Study

**DOI:** 10.3390/ijms24109073

**Published:** 2023-05-22

**Authors:** Paweł M. Potocki, Piotr Wójcik, Łukasz Chmura, Bartłomiej Goc, Marcin Fedewicz, Zofia Bielańska, Jakub Swadźba, Kamil Konopka, Łukasz Kwinta, Piotr J. Wysocki

**Affiliations:** 1Oncology Department, Faculty of Medicine, Jagiellonian University Medical College, 31-008 Cracow, Poland; 2Oncogene Diagnostics Sp. z o.o., 31-546 Cracow, Poland; 3Student Research Group, Oncology Department, Faculty of Medicine, Jagiellonian University Medical College, 31-008 Cracow, Poland; 4Radiotherapy Department, MSC National Research Institute of Oncology, 44-102 Gliwice, Poland; 5Józef Babiński Clinical Hospital, 30-393 Cracow, Poland; 6Klinik für Psychiatrie, Psychotherapie und Psychosomatik, Regio Klinikum, 25337 Elmshorn, Germany; 7Department of Laboratory Medicine, Faculty of Medicine and Health Sciences, Andrzej Frycz Modrzewski Krakow University, 30-705 Cracow, Poland

**Keywords:** metastatic colorectal cancer, *KRAS* mutation, *BRAF* mutation, V600E, G12C, tumour sidedness, brain metastases, MANEC, sotorasib, encorafenib

## Abstract

BRAF V600E and KRAS mutations that occur in colorectal cancer (CRC) define a subpopulation of patients with an inferior prognosis. Recently, the first BRAF V600E-targeting therapy has been approved and novel agents targeting KRAS G12C are being evaluated in CRC. A better understanding of the clinical characteristics of the populations defined by those mutations is needed. We created a retrospective database that collects clinical characteristics of patients with metastatic CRC evaluated for RAS and BRAF mutations in a single laboratory. A total of 7604 patients tested between October 2017 and December 2019 were included in the analysis. The prevalence of BRAF V600E was 6.77%. Female sex, primary in the right colon, high-grade, mucinous, signet cell, partially neuroendocrine histology, perineural and vascular invasion, and surgical tissue sample were factors associated with increased mutation rates. The prevalence of KRAS G12C was 3.11%. Cancer of primary origin in the left colon and in samples from brain metastases were associated with increased mutation rates. The high prevalence of the BRAF V600E mutation in cancers with a neuroendocrine component identifies a potential candidate population for BRAF inhibition. The association of KRAS G12C with the left part of the intestine and brain metastases of CRC are new findings and require further investigation.

## 1. Introduction

Metastatic colorectal cancer (CRC) is one of the leading causes of cancer-related mortality [1,2]. The mitogen-activated protein kinase (MAPK) cascade is the main pathway that transduces signalling from growth factor receptors. It is central in the regulation of proliferation, differentiation, and viability in healthy and malignant epithelial cells. Alterations increasing MAPK activity play a crucial role in cancer pathogenesis [3]. In addition to sustaining proliferative signalling, they are also closely related to the formation of metastases and drug resistance. Pathological MAPK activity is most commonly caused by activating mutations in genes in the *RAS* (Rat Sarcoma) family. Other pathomechanisms include epigenetic activation of growth factor receptors, activating mutations of genes for receptor tyrosine kinases, or activating mutations in signalling kinases downstream of RAS, i.e., RAF (rapidly accelerated fibrosarcoma) and MEK (mitogen-activated extracellular signal-related kinase) [3]. 

The *RAS* gene family consists of three isoforms, *KRAS*, *NRAS*, and *HRAS*, located on chromosomes 12, 1, and 11, respectively. They have a similar, evolutionarily conservative structure, consisting of four exons. RAS proteins play an important role in the regulation of multiple intracellular processes. By switching between their GTP-bound (Guanosine-5′-Triphosphate) active form and GDP-bound (Guanosine-5′-Diphosphate) inactive form, they control multiple signal cascades including those crucial to CRC pathogenesis: MAPK and PI3K (phosphatidylinositol 3-kinase). RAS itself can, in turn, be activated by multiple classes of signals including: by growth factor receptors through GRB2-SOS (Growth Factor Receptor-Bound Protein 2, Son of Sevenless) signal proteins; by intracellular AMP (Adenosine monophosphate) or calcium levels through GRF1 (Guanine Nucleotide Releasing Factor); by crosstalk with other signalling pathways through SHP2 (Src Homology Phosphatase 2) activity [3]. Activating mutations in *RAS* are found most frequently in pancreatic cancer (88.4% cases), colorectal cancer (55.1%), multiple myeloma (37.8%), lung adenocarcinoma (33.2%), follicular thyroid cancer (30.6%), cholangiocarcinoma (23.6%), endometrial cancer (20.0%), and skin melanoma (19.8%). In general, *KRAS* mutations occur more frequently than *HRAS* or *NRAS* mutations, although this is not true for all types of cancer [4]. 

In colorectal cancer, *KRAS* mutations occur in 50.4% of cases, *NRAS* mutations in 4.2%, and *HRAS* in 0.5% [4]. Their rate is higher in proximal colon tumours [5,6,7,8]. Hyperactive kinases resulting from mutated genes stimulate the MAPK pathway directly downstream of EGFR (Epithelial Growth Factor Receptor). Therefore, the *KRAS* and *NRAS* mutations are well-recognized negative predictors of EGFR inhibition. Their negative prognostic effect is also known, but due to a large heterogeneity of possible variations and the resulting kinase phenotypes, it is less well understood. 

Several large efforts aimed at developing RAS inhibitors failed to produce clinically active therapies; thus, for a long time, RAS has been considered an undruggable target. The difficulty has stemmed from many reasons, but mainly from the complexity of the RAS function: heterogeneity of its mutated isoforms, different modes of activation, presence of inter-pathway crosstalk, and feedback loops reactivating the blocked cascades [3,9]. Recently, new small-molecule agents that specifically block the G12C-mutated form of KRAS have emerged and are being clinically evaluated. These include Adagrasib (form. MRTX849, Mirati Therapeutics Inc., San Diego CA, USA), Sotorasib (form. AMG510, Amgen, Thousand Oaks CA, USA), D-1553 (MSD, Rahway, NJ, USA), GDC-6036 (Genentech Inc., South San Francisco, CA, USA), JAB-21822 (Jacobio Pharm., Beijing, China), JDQ443 (Novartis Pharm., Basel, Switzerland), JNJ-74699157 (Janssen R&D, Rarita, NJ, USA), and LY3499446 (Eli Lilly, Indianapolis, IN, USA). 

Within the MAPK cascade, RAF kinases transduce the signal from RAS, mainly through phosphorylative activation of downstream MEK kinases. Its three isoforms, ARAF, BRAF, and CRAF, occur in the inactive monomeric form. After activation, they bind to another RAF monomer, forming homodimers or heterodimers that exhibit kinase activity. The *BRAF* gene is located on chromosome 17 and can be affected by a variety of mutations. Those mutations typically occur in codon 600 (exon 15), substituting valine with another amino acid that results in a strongly hyperactive kinase. The most common allele is V600E. *BRAF* V600E mutations are commonly found in melanoma and lung, thyroid, and colorectal cancers, as well as gliomas and acute leukaemias [10]. Therapies targeting mutated BRAF are now a standard of care in a subset of patients with melanoma and lung cancer. 

In colorectal cancer, the *BRAF* V600E mutation is associated with a distinct clinical and pathological phenotype: predominantly proximal colon tumour location, aggressive growth, adverse metastasizing patterns, resistance to the EGFR blockade, and an overall survival (OS) rate that is 1.5 to 3 times shorter as compared to the general population. Naturally, it has been investigated as a potential therapeutic target. Unlike the high efficacy of BRAF targeting strategies observed in melanoma, monotherapy with V600E-specific kinase inhibitors (vemurafenib, dabrafenib) has been proven to be ineffective in metastatic colorectal cancer (mCRC) [11]. It has been shown to trigger the feedback loop that facilitates alternative MAPK activation mechanisms, particularly through increased EGFR and MEK activity [11]. Simultaneous inhibition of this feedback loop through inhibition of EGFR or MEK has been shown to suppress the feedback, thus restoring control over BRAF V600E-activated MAPK [12]. The efficacy of such a strategy was confirmed in clinical trials, most recently in a phase III BEACON study that demonstrated the feasibility of a combined BRAF blockade in a randomized setting. The study also demonstrated that triple inhibition (anti-EGFR + anti-BRAF V600E + anti-MEK with cetuximab, encorafenib, binimetinib) appears to be not more effective than double inhibition (anti-EGFR + anti-BRAF V600E with cetuximab, encorafenib) [13]. As a result, the cetuximab-encorafenib combination is now a standard of care in patients with pretreated *BRAF* V600E-mutated mCRC. Trials evaluating the first-line efficacy, alternative, and derivative combinations are also under way. Noticeably, another double inhibition strategy (anti-BRAF V600E + anti-MEK) was subjected to an early clinical evaluation [14]. Agents with anti-BRAF activity that have been investigated in this setting include Vemurafenib (Genentech/Roche), Dabrafenib (GlaxoSmithKline), Encorafenib (Array BioPharma/Pierre Fabre), Regorafenib (Bayer), Sorafenib (Bayer), Lifirafenib (BeiGene), Agerafenib (Roche), Belvarafenib (Hanmi Pharmaceutical), Ro 5126766 (Chugai Pharmaceutical), and LY3009120 (Eli Lilly). 

Antibodies targeting EGFR (cetuximab and panitumumab) have transformed systemic treatment for advanced CRC and are now an integral part of first-line regimens. As a result, molecular testing for *KRAS, NRAS* and *BRAF* mutations has become an obligatory part of the standard diagnostic routine. As MAPK pathway deregulation in CRC becomes better understood, novel, effective therapies targeting this signalling cascade, in unique, molecularly-defined mCRC patient subpopulations, are emerging. The aim of this study is to define the clinical characteristics of mCRC patients harbouring readily targetable mutations (KRAS G12C or BRAF V600E) and to screen for potential clinical factors predictive of the occurrence of the mutations. 

## 2. Results

### 2.1. Population Characteristics

A total of 7798 cases tested between October 2017 and December 2019 were identified. After discarding 89 duplicates, 102 nondiagnostic results and 3 non-colorectal cancers, 7604 were included in the analysis (Figure 1). Men made up 58.8%. A total of 75.5% were over 60 and 4.91% were over 80 years old. The distal primary location was more prevalent, with the rectum being the most common one. The majority of the analysed samples originated in the primary tumour (87.9%). Of the 685 samples not originating from primaries, the most common sample source sites were the liver, peritoneal implants, and local recurrence. Surgical samples comprised the majority of the cases (73.49%), with endoscopic biopsy being the second most common sampling method (22.91%). The BRAF V600E mutation occurred in 515 cases (6.77%), KRAS G12C in 237 (3.12%), and any other KRAS in 3591 (47.23%). The detailed characteristics of the general population and the subpopulations are outlined in Table 1 and Appendix A. 

### 2.2. BRAF V600E Mutation

The prevalence of *BRAF* V600E was 6.77%. The characteristics of mutated cases compared to those without mutations are outlined in Table 2. *BRAF* V600E-mutated cancers exhibited higher T and N scores, as well as higher rates of vascular and perineural invasion compared to non-mutated ones. 

The mutation was present more often in women (odds ratio (OR) 2.009, 95% confidence interval (CI) 1.677–2.408, *p* < 0.001). A strong yet nonsignificant trend towards greater mutation risk in older age was observed (Figure 2). The mutation was more likely to be found in high-grade cancers (OR 4.061, 95% CI 3.218–5.125, *p* < 0.001). It was more likely to occur in cancers with mucinous histology (OR 3.430, 95% CI 1.415–8.315, *p* = 0.006) or partially mucinous histology (OR 2.718, 95% CI 2.121–3.482, *p* < 0.001) or with the presence of signet cells (OR 1.544, 95% CI 1.104–2.160, *p* = 0.011). Mixed adeno-neuroendocrine cancers (MANEC) were more likely to carry the mutation (OR 9.211, 95% CI 3.071–27.626, *p* < 0.001). Proximal tumours, occurring in the caecum, ascending colon, hepatic flexure, or transverse colon, were more likely to carry the mutation as compared to distal ones occurring in the splenic flexure, descending colon, sigmoid colon, rectosigmoid junction, or rectum (OR 8.356, 95% CI 6.732–10.372, *p* < 0.001). Rectal/rectosigmoid tumours were less likely to carry the mutation as compared to the abdominal (proximal and distal combined) part of the colon (OR 0.231, 95% CI 0.179–0.299, *p* < 0.001), although the difference between rectal/rectosigmoid and distal abdominal tumours (occurring in splenic flexure or descending and sigmoid colon) was not significant (OR 1.32, 95% CI 0.928–1.876, *p* = 0.122). There was no significant difference in the mutation rate between primary, locally recurrent, and metastatic lesions, although a trend towards a higher rate was observed in primaries. The mutation was less likely to occur in samples from biopsies as compared with surgical ones. All samples secured during colonoscopy were 30.9% less likely to produce a positive result as compared to all samples secured during surgeries (OR 0.691, 95% CI 0.532–0.897, *p* = 0.006). All samples secured during needle biopsies were 58.7% less likely to produce a positive result as compared to all samples secured during surgeries (OR 0.691, 95% CI 0.532–0.897, *p* = 0.023). The same was true when restricted to metastatic lesions only. A biopsy was 57.1% less likely to produce a BRAF V600E-mutated sample than a surgery (OR 0.429, 95% CI 0.202–0.911, *p* = 0.028). Variability in the rate of BRAF V600E mutation was also observed according to geographical regions. Only the difference between the regions with the highest sample numbers (Lodzkie < Slaskie) was statistically significant (OR 0.669, 95% CI 0.481–0.930, *p* = 0.017). (Table 3 and Appendix A, Figure 2 and Figure 3).

### 2.3. KRAS G12C Mutation

The prevalence of *KRAS* G12C in the overall population was 3.12% (6.60% of all *KRAS* mutations). The pathomorphological characteristics of the cases with the *KRAS* G12C mutation compared to the rest of the cohort and other *KRAS*-mutated cases are outlined in Table 4. 

When comparing *KRAS* G12C mutant cases with the rest of the cohort, proximal tumours were less likely to carry the *KRAS* G12C mutation compared to distal tumours (OR 0.652, 95% CI 0.451–0.944, *p* = 0.024). Rectal/rectosigmoid tumours were more likely to carry the mutation as compared to those of the abdominal part of the colon (OR 1.470, 95% CI 1.122–1.925, *p* = 0.005), although the difference between the rectal/rectosigmoid and intraabdominal distal colon was borderline insignificant (OR 0.729, 95% CI 0.521–1.021, *p* = 0.066). CNS metastases were 21 times more likely to carry the G12C mutation than the rest of the cohort (OR 20.98, 95% CI 3.48–126.47, *p* = 0.001) (Table 5 and Appendix A, Figure 4).

When comparing *KRAS* G12C-mutant cases with the rest of the *KRAS*-mutant cases, proximal tumours were less likely to carry the KRAS G12C mutation compared to distal tumours (OR 0.618, 95% CI 0.425–0.898, *p* = 0.012), whereas rectal/rectosigmoid tumours were more likely to carry the mutation as compared to the abdominal part of the colon (OR 1.425, 95% CI 1.083–1.874, *p* = 0.011). CNS metastases were 15.5 times more likely to carry the G12C mutation than the rest of the cohort (OR 15.529, 95% CI 2.173–110.997, *p* = 0.006). Higher rates of G12C mutation were observed in samples taken from needle biopsies but not endoscopic biopsies (OR 2.05, 95% CI 1.096–3.833, *p* = 0.025). This finding was not repeated when restricting the comparison to only metastatic cases and is not present when comparing G12C cases with the general population. (Table 6 and Appendix A).

## 3. Discussion

Our study used one of the largest single-laboratory-generated databases comprising cases of advanced colorectal Caucasian patients worldwide. In the recent meta-analysis of 67 studies (32,407 patients) evaluating the clinical and pathological significance of the *BRAF* V600E mutation in colorectal cancer, the largest study collected less than 2000 cases. Our analysis comprises 7604 patients’ records, and this population would represent approximately 23% of all summarized records in the aforementioned meta-analysis. Moreover, there is not a single large-scale study published to date evaluating the clinicopathological features associated with *KRAS* G12C mutation, which is expected to shortly become the first clinically targeted *KRAS* mutation in CRC. 

The prevalence of *BRAF* V600E mutations (below 7% (6.77%)) is lower than expected, based on previously published data. The observed mutation rate varies considerably when compared across other studied cohorts. The aforementioned meta-analysis (n = 32,407) reported a mean rate of 11.34% (3.14–23.14%) [15]. In contrast, previous, much smaller studies conducted on the Polish population showed values similar to the one reported in our analysis. A study on non-metastatic CRC patients found the mutation in 2/163 (1.22%) cases [16]. Two Polish studies on advanced CRC reported the incidence of BRAF V600E mutations in 24/500 (4.80%) and 7/102 (6.86%), respectively [17,18]. A recently published large study on the Russian population reported the incidence at 6.7% [19]. The observed variability may be due to a number of reasons, described below. However, it must be underscored that our analysis was unique in its homogeneity, in that all of the tested patients were Caucasian. 

Most studies found that the *BRAF* V600E mutation was more likely to occur in women and in proximal tumours, which is also a finding in the current study. An association of mutation risk with age has also been described, although it did not reach the significance threshold in this study [20]. In some studies, a higher clinical stage at the time of diagnosis was associated with an increased risk of mutation [15,21,22]. The distribution of these and other risk factors among different cohorts may explain the variability. 

Another reason for the lower-than-expected *BRAF* V600E mutation rates may be survivor bias. The V600E mutation is associated with an aggressive phenotype, adverse metastasizing patterns, and rapid progression. Therefore, it has been shown to negatively impact OS in patients with radically treated and metastatic colorectal cancer. Interestingly, the same studies did not show a significant impact on disease-free survival (DFS) or progression-free survival (PFS) [23,24,25,26,27]. Therefore, it can be concluded that it is the fast rate of disease progression that drives the decrease in OS, whereas the effectiveness of treatment (surgical or systemic as measured in recurrence rates and PFS) does not appear to be affected by the mutation. This indirectly hints that the mutation rate in the radically treated population should remain unchanged in the subpopulation of patients who ultimately recur. In this cohort, patients were only tested for the *BRAF* mutation when presenting with metastatic disease. Effectively, only patients who maintained a good general condition for long enough to be diagnosed, tested, and treated were included in the analysis. Therefore, subpopulations with adverse prognostic factors for metastatic disease (including *BRAF* V600E mut, older age, higher tumour burden, poor performance status) are likely to be underrepresented. This selection practice resulted directly from the national reimbursement policy. The authors are aware that most clinical practice guidelines for CRC treatment recommend testing at the time of first diagnosis.

In this study, the samples secured during surgery contained significantly higher numbers of BRAF V600E mutations than the samples secured by endoscopy. A similar relationship was not observed for the KRAS G12C mutation. Therefore, in the authors’ opinion, it is unlikely that the BRAF V600E mutation rate was affected by sample processing or the methodology for detecting mutations. The difference in the BRAF V600E rate between surgical and endoscopic samples may represent the differences in the possibilities of collecting material during colonoscopy. It is not always possible to reach the right side of the colon with the endoscope; therefore, the material from this location is less often collected during endoscopic examination. The BRAF-mutated tumours are associated with a right-sided location in the colon and such samples are less likely to be secured by endoscopy.

Tumour histology is another factor that was associated with the appearance of the *BRAF* mutation. This study confirmed the well-documented link between the risk of the *BRAF* mutation and mucinous histology, high tumour grade, and vascular and perineural invasion. The presence of signet cells or partial neuroendocrine components also increased the odds of the mutation. The last finding, although previously described [28], suggests a potential therapeutic target in this rare and clinically difficult cancer subtype. There are already case studies describing responses to dual inhibition of BRAF and MEK in this population [29,30]. Recently, a dabrafenib-trametinib combination has gained tissue-agnostic approval in the United States [31], which now allows any malignancy with the mutation to be treated with this regimen. 

Tumour sidedness has also been shown to affect the rate of *BRAF* mutation in this study, with proximal (right-sided) tumours manifesting higher rates. Notably, rectal/rectosigmoid tumours were associated with significantly lower mutation rates than tumours located in the abdominal part of the colon. Because the occurrence of the mutation did not differ significantly between the rectal/rectosigmoid and distal abdominal colon (only a moderate trend was present), this difference is likely a manifestation of well-documented biological differences between the proximal and distal parts of the large intestine. A study on a larger dataset may resolve this doubt. This study also confirms the link between the risk of *BRAF* mutation and female sex, mucinous histology, high tumour grade, vascular and perineural invasion, and high T and N scores, which is also well documented in the literature [32]. 

In this study, some geographical variability is evidenced in the rates of BRAF mutations. Only the difference between the Śląskie and Łódzkie regions is significant, but notably, those were the ones with the highest number of samples. Nonsignificant trends for other regions are also present. In the authors’ opinion, this finding should be further evaluated in a larger study. This may provide insight into hereditary and environmental factors that influence the rates of BRAF mutations, but it may also be a manifestation of practice patterns that influenced patient selection for this cohort. 

The frequency of KRAS G12C mutations in this study was 3.12%, which is similar to recently published datasets [33,34,35]. Surprisingly, the *KRAS* G12C mutation in the Polish population occurred more frequently in distal tumours, contrary to a known predisposition of *RAS* mutations for the proximal part of the colon [5,6,7,8] and data on *KRAS* G12C-mutated tumours in the Scandinavian population [35]. Additionally, samples from rectal/rectosigmoid cancers were more likely to carry the mutation compared to samples from the intraabdominal part of the intestine. These findings were true both in comparison to the rest of the cohort and in comparison to the rest of the *KRAS*-mutated cases. Given that there was only a borderline nonsignificant trend towards a higher mutation rate in the rectal/rectosigmoid tumours as compared to the left colon tumours, this should be interpreted with caution, as it may only be a manifestation of the aforementioned predisposition to the distal parts of the intestine. This predisposition for left-sidedness seems to be a new characteristic of the G12C mutation and should be confirmed in other datasets. The authors identified four recent publications that included sidedness in *KRAS* G12C vs. *KRAS* non-G12C population characteristics. In two of them, there was a trend towards a higher percentage of left-sided primaries in the *KRAS* G12C cohort compared to the cohort with the remaining *RAS* mutations [35,36]. In the remaining two, a higher percentage of right-sided primaries was reported in *KRAS* G12C-mutated cancers [33,37]. The aforementioned trends were not statistically significant. 

Of the five samples originating from brain metastases in this study, two are *KRAS* G12C-mutated. Although the numbers are small, the rates of G12C occurrence in brain metastases differ significantly, both compared to the rest of the cohort and to the rest of the *KRAS*-mutated cases. Notably, another two of the brain metastasis samples harboured *KRAS* mutations (G12D and G13D). Evidence that *KRAS* mutations in general may play a role in the pathogenesis of CRC brain metastases has previously been described [38]. The finding in this study suggests that there may be a link between brain metastases and the G12C mutation, specifically. Recent studies involving brain metastases from lung cancer have reported similar findings [39,40]. If confirmed in further studies, the overrepresentation of the G12C mutation in brain metastases may be of clinical significance, in light of recent preliminary data on sotorasib being active in brain metastases [41]. 

A higher prevalence of *KRAS* G12C mutations in samples taken from needle biopsies is a puzzling finding. It only presented itself in the comparison between G12C and other *KRAS* mutations. A similar finding was not found for endoscopic biopsies and when the analysis was restricted to metastatic samples only. This may be a manifestation of selection bias, but the exact mechanisms are unclear and require further investigation. 

## 4. Materials and Methods

We created a dedicated retrospective database of consecutive patients with metastatic colorectal cancer from 107 Polish institutions whose tissue samples had been tested for *KRAS*, *NRAS,* and *BRAF* mutations in a single laboratory. The cohort consisted of samples that were tested as part of standard practice. Although there was no active selection strategy, patients were passively selected, as testing was driven by the restrictive inclusion criteria of a national reimbursement policy. Clinical data were extracted from available records, including basic demographics (age, sex, geographic region), TNM scores (8th edition, 2016), histology (NOS, mucinous, partially mucinous, signet cell, cribriform, comedo-like, partially neuroendocrine, other), primary location (right or proximal colon including caecum, ascending colon, hepatic flexure and transverse colon; left or distal colon including splenic flexure, descending and sigmoid colon; rectum/rectosigmoid junction), sample origin (primary, local recurrence, metastatic, according to location of metastases), and sampling method (endoscopic biopsy, needle biopsy, surgery). 

The suitability of the samples (amount of tissue, neoplastic cells content) was assessed by a pathologist. The appropriate tissue fragments were collected from microscopic slides by macrodissection. The minimum requirement for analysis was 20% of neoplastic cells, and in most cases (90% of samples) the percentage of neoplastic cells exceeded 50%. A validated Sanger sequencing technique with a detection limit of 10% of the mutant DNA (approx. 20% of cells) was used to test the mutations. The mutation analysis included selected gene fragments: KRAS exon 2 (including codons 10–22), KRAS exon 3 (including codons 58–64), KRAS exon 4 (including codons 117, 146), NRAS exon 2 (including codons 10–18), NRAS exon 3 (including codons 58–64), NRAS exon 4 (including codons 117, 146), BRAF exon 15 (including codons 594–601). The appropriate DNA fragments were amplified by the polymerase chain reaction and subsequently sequenced with the BigDye Terminator v3.1 Cycle Sequencing Kit (Thermo Fisher Scientific) on a sequencer (3730 DNA Analyzer, Applied Biosystems). In parallel to the main Sanger sequencing technique, an independent screening method (SSCP or HRM) was used as an intra-laboratory control in each case. The mutational analysis was performed by a single laboratory that successfully participates in external quality assessment trials organized by ESP and GenQA. 

Statistical analysis was performed in the R environment v 4.1.3 [42]. Qualitative variables were compared between groups using the chi-square test (with Yates correction for 2 × 2 tables). Fisher’s exact test was used when low expected quantities occurred in the tables. Quantitative variables were compared between groups using the Mann–Whitney test. A *p*-value of 0.05 was considered statistically significant.

## 5. Conclusions

Our nationwide study clearly demonstrates that BRAF V600E and KRAS G12C colorectal tumours show significantly detrimental clinicopathological features. On the other hand, the same mutations are highly predictive of the clinical benefit of novel targeted therapies.

The characteristics of the population with *BRAF* V600E-mutated CRC were generally consistent with the literature, apart from the lower overall prevalence of the mutation, the anomalies in its geographical distribution, and differences between sample types. These require further investigation. The increased prevalence of the *BRAF* V600E mutation in partially neuroendocrine cancers may identify a new subpopulation eligible for BRAF inhibition. The characteristics of the *KRAS* G12C mutation revealed two potentially novel findings: the higher prevalence of the *KRAS* G12C mutation in distal (left-sided tumours) contrary to other KRAS mutations and the higher prevalence of the *KRAS* G12C mutation in brain metastases compared to other KRAS mutations. Those require confirmation in other studies but may become clinically relevant with the ongoing development of specific *KRAS* G12C inhibitors. 

With the rise of therapies targeting signal transduction in pathways central to CRC pathogenesis, new subpopulations defined by the ever-growing number of molecular biomarkers emerge. The clinical characterization of these subpopulations is both a requirement for the introduction of new drugs into clinical practice and a source of hypotheses that may affect the direction of future research. The authors present the first results from a newly created database, which hopefully will be expanded with more cases, survival data, and biomarkers to fuel future research. 

## Figures and Tables

**Figure 1 ijms-24-09073-f001:**
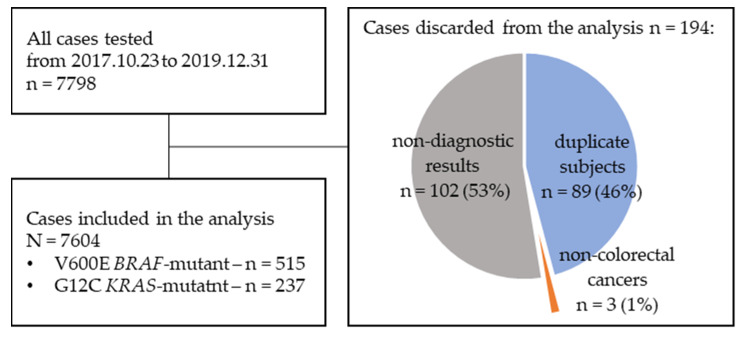
Data selection flowchart.

**Figure 2 ijms-24-09073-f002:**
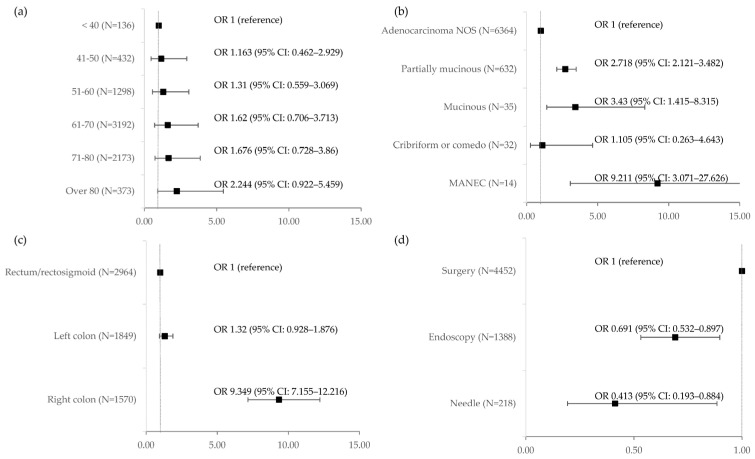
Odds ratios for BRAF V600E mutation incidence according to selected factors: (**a**) According to patient’s age; (**b**) According to tumour histological type; (**c**) According to tumour location in the large intestine; (**d**) According to procedure resulting in the assessed sample.

**Figure 3 ijms-24-09073-f003:**
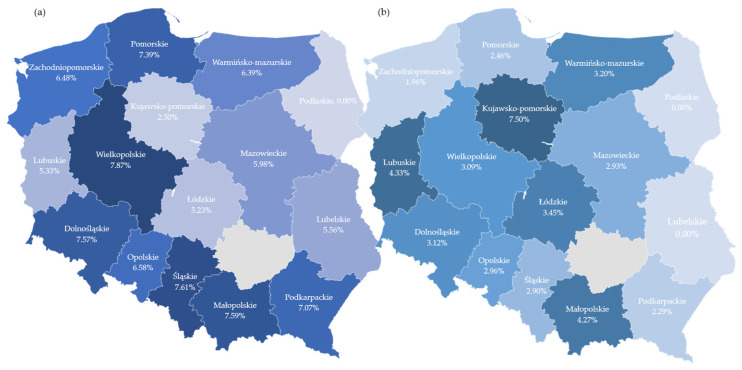
Geographical distribution of the incidence of (**a**) BRAF V600E and (**b**) KRAS G12C mutations. The lighter the colour the lower the mutation incidence.

**Figure 4 ijms-24-09073-f004:**
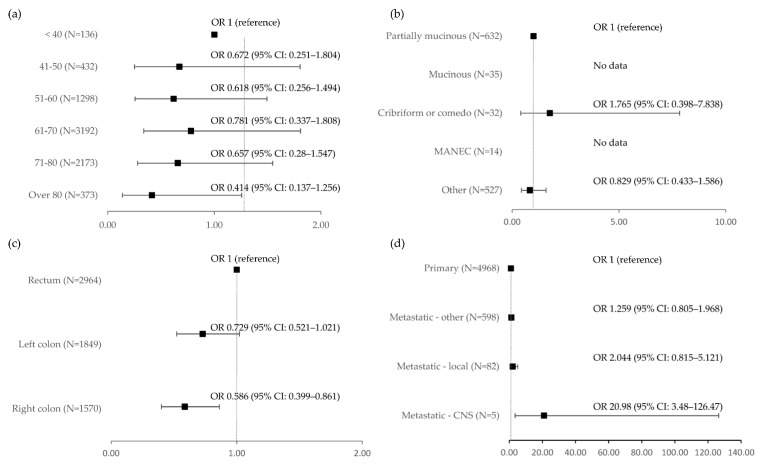
Odds ratios for KRAS G12C mutation incidence according to selected factors: (**a**) According to patient’s age; (**b**) According to tumour histological type; (**c**) According to tumour location in the large intestine; (**d**) According to type of cancer lesion.

**Table 1 ijms-24-09073-t001:** Clinical and pathological characteristics of the study population according to mutation status.

Parameter		Populations Defined by Mutation Types			
		Overall	BRAF V600E	KRAS G12C	Other KRAS
		(N = 7604)	(N = 515)	(N = 237)	(N = 3591)
Sex	Female	3132 (41.19%)	295 (57.28%)	101 (42.62%)	1527 (42.52%)
	Male	4471 (58.81%)	220 (42.72%)	136 (57.38%)	2064 (57.48%)
Age (years)	<30	21 (0.28%)	0 (0.00%)	0 (0.00%)	10 (0.28%)
	31–40	115 (1.51%)	6 (1.17%)	6 (2.53%)	49 (1.36%)
	41–50	432 (5.68%)	22 (4.27%)	13 (5.49%)	190 (5.29%)
	51–60	1298 (17.07%)	74 (14.37%)	36 (15.19%)	588 (16.37%)
	61–70	3192 (41.98%)	222 (43.11%)	111 (46.84%)	1486 (41.38%)
	71–80	2173 (28.58%)	156 (30.29%)	64 (27.00%)	1070 (29.80%)
	over 80	373 (4.91%)	35 (6.80%)	7 (2.95%)	198 (5.51%)
Tumour stage	T1	39 (0.98%)	0 (0.00%)	1 (0.83%)	25 (1.29%)
	T2	291 (7.29%)	10 (3.33%)	11 (9.09%)	144 (7.44%)
	T3	2611 (65.37%)	175 (58.33%)	81 (66.94%)	1243 (64.20%)
	T4	1039 (26.01%)	115 (38.33%)	28 (23.14%)	516 (26.65%)
	Tis	14 (0.35%)	0 (0.00%)	0 (0.00%)	8 (0.41%)
Nodal stage (0 vs. 1 vs. 2)	N0	1006 (26.18%)	57 (19.79%)	37 (31.09%)	486 (26.00%)
	N1	1460 (37.99%)	94 (32.64%)	43 (36.13%)	740 (39.59%)
	N2	1377 (35.83%)	137 (47.57%)	39 (32.77%)	643 (34.40%)
Histology	Adenocarcinoma NOS	6364 (89.93%)	362 (78.02%)	196 (88.69%)	3108 (88.52%)
	Partially mucinous	632 (8.93%)	89 (58.17%)	23 (56.10%)	382 (56.26%)
	Mucinous	35 (0.49%)	6 (3.92%)	0 (0.00%)	10 (1.47%)
	Cribriform or comedo	32 (0.45%)	2 (1.31%)	2 (4.88%)	9 (1.33%)
	MANEC	14 (0.2%)	5 (3.27%)	0 (0.00%)	2 (0.29%)
Grade	G1	590 (12.14%)	26 (7.60%)	17 (10.90%)	299 (12.88%)
	G2	3509 (72.22%)	181 (52.92%)	122 (78.21%)	1708 (73.59%)
	G3	756 (15.56%)	134 (39.18%)	17 (10.90%)	313 (13.49%)
	G4	4 (0.08%)	1 (0.29%)	0 (0.00%)	1 (0.04%)
Primary tumour sidedness	Left colon or rectum	4827 (75.46%)	129 (30.57%)	163 (82.32%)	2255 (74.20%)
	Right colon	1570 (24.54%)	293 (69.43%)	35 (17.68%)	784 (25.80%)
Sample type	Endoscopic biopsy	1388 (22.91%)	73 (17.76%)	45 (22.84%)	654 (22.58%)
	Needle biopsy	218 (3.6%)	7 (1.70%)	12 (6.09%)	90 (3.11%)
	Surgery	4452 (73.49%)	331 (80.54%)	140 (71.07%)	2152 (74.31%)
Sample source	Primary	4968 (87.88%)	346 (90.58%)	153 (83.61%)	2376 (88.43%)
	Metastatic	685 (12.12%)	36 (9.42%)	30 (16.39%)	311 (11.57%)
Location of metastatic samples	Local recurrence	81 (11.82%)	3 (0.79%)	5 (2.73%)	39 (1.45%)
	Liver	266 (38.83%)	11 (2.88%)	11 (6.01%)	108 (4.02%)
	Lung	73 (10.66%)	3 (0.79%)	1 (0.55%)	42 (1.56%)
	Peritoneal	148 (21.61%)	12 (3.14%)	5 (2.73%)	76 (2.83%)
	Nodal	31 (4.53%)	3 (0.79%)	2 (1.09%)	10 (0.37%)
	Ovarian	33 (4.82%)	2 (0.52%)	1 (0.55%)	10 (0.37%)
	Small intestine	26 (3.8%)	2 (0.52%)	2 (1.09%)	12 (0.45%)
	CNS	5 (0.73%)	0 (0%)	2 (1.09%)	2 (0.07%)
	Skin/subcutaneous	22 (3.21%)	0 (0%)	1 (0.55%)	12 (0.45%)

Expanded version available in the (Appendix A). Right colon (proximal) included caecum, ascending colon, hepatic flexure, and transverse colon. Left colon (distal) included splenic flexure and descending and sigmoid colon. CNS—central nervous system; MANEC—mixed adeno-neuroendocrine cancer; NOS—not otherwise specified.

**Table 2 ijms-24-09073-t002:** Pathomorphological characteristics of primary tumours in *BRAF* V600E-mutated cancers.

Parameter		Group		*p*
		BRAF V600E Mutation (N = 515)	No BRAF V600E Mutation (N = 7089)	
Tumour stage	T1	0 (0.00%)	39 (1.06%)	*p* < 0.001 *
	T2	10 (3.33%)	281 (7.61%)	
	T3	175 (58.33%)	2436 (65.94%)	
	T4	115 (38.33%)	924 (25.01%)	
	Tis	0 (0.00%)	14 (0.38%)	
Nodal stage	N0	57 (19.32%)	949 (25.91%)	*p* < 0.001 *
	N1a	40 (13.56%)	476 (12.99%)	
	N1b	43 (14.58%)	692 (18.89%)	
	N1c	11 (3.73%)	198 (5.41%)	
	N2a	55 (18.64%)	594 (16.22%)	
	N2b	82 (27.80%)	646 (17.64%)	
	Nx	7 (2.37%)	108 (2.95%)	
Nodal stage (0 vs. 1 vs. 2)	N0	57 (19.79%)	949 (26.69%)	*p* < 0.001 *
	N1	94 (32.64%)	1366 (38.42%)	
	N2	137 (47.57%)	1240 (34.88%)	
Nodal stage (positive vs. negative)	Negative	57 (19.79%)	949 (26.69%)	*p* = 0.013 *
	Positive	231 (80.21%)	2606 (73.31%)	
Angioinvasion	No	41 (21.69%)	775 (34.63%)	*p* < 0.001 *
	Yes	148 (78.31%)	1463 (65.37%)	
Perineural invasion	No	44 (38.94%)	696 (51.98%)	*p* = 0.01 *
	Yes	69 (61.06%)	643 (48.02%)	
Biopsy vs. surgery for primary samples	Biopsy	80 (19.46%)	1526 (27.02%)	*p* = 0.001 *
	Surgery	331 (80.54%)	4121 (72.98%)	

Right colon (proximal) included caecum, ascending colon, hepatic flexure, and transverse colon. Left colon (distal) included splenic flexure and descending and sigmoid colon. Rectum included rectosigmoid tumours. MANEC—mixed adeno-neuroendocrine cancer. *p*—for quantitative variables, Mann–Whitney test; for qualitative variables, chi-squared test or exact Fisher test. *—statistically significant difference (*p* < 0.05).

**Table 3 ijms-24-09073-t003:** Clinical and pathological factors that affected the rate of BRAF V600E mutation.

Trait	Group	BRAF V600E	OR	95% CI		*p*
Sex	Male (N = 4471)	220 (4.92%)	1	ref.		
	Female (N = 3132)	295 (9.42%)	2.009	1.677	2.408	<0.001
Age (years)	<40 (N = 136)	6 (4.41%)	1	ref.		
	41–50 (N = 432)	22 (5.09%)	1.163	0.462	2.929	0.749
	51–60 (N = 1298)	74 (5.70%)	1.31	0.559	3.069	0.534
	61–70 (N = 3192)	222 (6.95%)	1.62	0.706	3.713	0.255
	71–80 (N = 2173)	156 (7.18%)	1.676	0.728	3.86	0.225
	Over 80 (N = 373)	35 (9.38%)	2.244	0.922	5.459	0.075
Histology	Adenocarcinoma NOS (N = 6364)	362 (5.69%)	1	ref.		
	Partially mucinous (N = 632)	89 (14.08%)	2.718	2.121	3.482	<0.001
	Mucinous (N = 35)	6 (17.14%)	3.43	1.415	8.315	0.006
	Cribriform or comedo (N = 32)	2 (6.25%)	1.105	0.263	4.643	0.891
	MANEC (N = 14)	5 (35.71%)	9.211	3.071	27.626	<0.001
Mucous component	No (N = 6555)	385 (5.87%)	1	ref.		
	Yes (N = 1049)	130 (12.39%)	2.267	1.837	2.798	<0.001
Signet cells presence	No (N = 7187)	474 (6.60%)	1	ref.		
	Yes (N = 417)	41 (9.83%)	1.544	1.104	2.16	0.011
Grade	G1 (N = 590)	26 (4.41%)	1	ref.		
	G2 (N = 3509)	181 (5.16%)	1.18	0.775	1.797	0.441
	G3 (N = 756)	134 (17.72%)	4.673	3.024	7.221	<0.001
	G4 (N = 4)	1 (25.00%)	7.231	0.727	71.909	0.091
Primary tumour localization	Rectum/rectosigmoid (N = 2964)	71 (2.40%)	1	ref.		
	Left colon (N = 1849)	58 (3.14%)	1.32	0.928	1.876	0.122
	Right colon (N = 1570)	293 (18.66%)	9.349	7.155	12.216	<0.001
Sample type	Surgery (N = 4452)	331 (7.43%)	1	ref.		
	Endoscopy (N = 1388)	73 (5.26%)	0.691	0.532	0.897	0.006
	Needle (N = 218)	7 (3.21%)	0.413	0.193	0.884	0.023
Sample origin	Primary (N = 4968)	346 (6.96%)	1	ref.		
	Metastatic (N = 685)	36 (5.26%)	0.741	0.521	1.055	0.096
Biopsy vs. surgery for metastatic tissues	Surgery (N = 854)	59 (6.91%)	1	ref.		
	Biopsy (N = 259)	8 (3.09%)	0.429	0.202	0.911	0.028

Right colon (proximal) included caecum, ascending colon, hepatic flexure, and transverse colon. Left colon (distal) included splenic flexure and descending and sigmoid colon. Rectum included rectosigmoid tumours. MANEC—mixed adeno-neuroendocrine cancer. *p*—univariate logistic regressions. Expanded version available in the (Appendix A).

**Table 4 ijms-24-09073-t004:** Pathomorphological characteristics of primary tumours in *KRAS* G12C-mutated cancers.

Parameter		Group				
		KRAS G12C (N = 237)	No KRAS G12C (N = 7367)	*p*	Other KRAS (N = 3591)	*p*
Tumour stage	T1	1 (0.83%)	38 (0.98%)	*p* = 0.835		*p* = 0.859
	T2	11 (9.09%)	280 (7.23%)		144 (7.44%)	
	T3	81 (66.94%)	2530 (65.32%)		1243 (64.20%)	
	T4	28 (23.14%)	1011 (26.10%)		516 (26.65%)	
	Tis	0 (0.00%)	14 (0.36%)		8 (0.41%)	
Nodal stage	N0	37 (30.33%)	969 (25.26%)	*p* = 0.051	486 (25.34%)	*p* = 0.062
	N1a	8 (6.56%)	508 (13.24%)		272 (14.18%)	
	N1b	27 (22.13%)	708 (18.46%)		354 (18.46%)	
	N1c	8 (6.56%)	201 (5.24%)		114 (5.94%)	
	N2a	26 (21.31%)	623 (16.24%)		307 (16.01%)	
	N2b	13 (10.66%)	715 (18.64%)		336 (17.52%)	
	Nx	3 (2.46%)	112 (2.92%)		49 (2.55%)	
Nodal stage (0 vs. 1 vs. 2)	N0	37 (31.09%)	969 (26.02%)	*p* = 0.457	486 (26.00%)	*p* = 0.466
	N1	43 (36.13%)	1417 (38.05%)		740 (39.59%)	
	N2	39 (32.77%)	1338 (35.93%)		643 (34.40%)	
Nodal stage (positive vs. negative)	Negative	37 (31.09%)	969 (26.02%)	*p* = 0.257	486 (26.00%)	*p* = 0.265
	Positive	82 (68.91%)	2755 (73.98%)		1383 (74.00%)	
Angioinvasion	No	26 (30.95%)	790 (33.72%)	*p* = 0.682	434 (36.94%)	*p* = 0.326
	Yes	58 (69.05%)	1553 (66.28%)		741 (63.06%)	
Perineural invasion	No	25 (46.30%)	715 (51.14%)	*p* = 0.575	375 (53.19%)	*p* = 0.403
	Yes	29 (53.70%)	683 (48.86%)		330 (46.81%)	
Biopsy vs. surgery for primary samples	Biopsy	57 (28.93%)	1549 (26.43%)	*p* = 0.483	744 (25.69%)	*p* = 0.357
	Surgery	140 (71.07%)	4312 (73.57%)		2152 (74.31%)	

Right colon (proximal) included caecum, ascending colon, hepatic flexure, and transverse colon. Left colon (distal) included splenic flexure and descending and sigmoid colon. Rectum included rectosigmoid tumours. MANEC—mixed adeno-neuroendocrine cancer. *p*—for quantitative variables, Mann–Whitney test; for qualitative variables, chi-squared test or exact Fisher test. Expanded version available in the (Appendix A).

**Table 5 ijms-24-09073-t005:** Clinical and pathological factors that affected the rate of *KRAS* G12C mutation.

Trait	Group	KRAS G12C	OR	95% CI		*p*
Sex	Male (N = 4471)	136 (3.04%)	1	ref.		
	Female (N = 3132)	101 (3.22%)	1.062	0.818	1.38	0.651
Age (years)	<40 (N = 136)	6 (4.41%)	1	ref.		
	41–50 (N = 432)	13 (3.01%)	0.672	0.251	1.804	0.43
	51–60 (N = 1298)	36 (2.77%)	0.618	0.256	1.494	0.285
	61–70 (N = 3192)	111 (3.48%)	0.781	0.337	1.808	0.563
	71–80 (N = 2173)	64 (2.95%)	0.657	0.28	1.547	0.337
	Over 80 (N = 373)	7 (1.88%)	0.414	0.137	1.256	0.119
Histology	Partially mucinous (N = 632)	23 (3.64%)	1	ref.		
	Mucinous (N = 35)	0 (0.00%)	--	--	--	--
	Cribriform or comedo (N = 32)	2 (6.25%)	1.765	0.398	7.838	0.455
	MANEC (N = 14)	0 (0.00%)	--	--	--	--
	Other (N = 527)	16 (3.04%)	0.829	0.433	1.586	0.571
Primary tumour localization	Rectum (N = 2964)	111 (3.74%)	1	ref.		
	Left colon (N = 1849)	51 (2.76%)	0.729	0.521	1.021	0.066
	Right colon (N = 1570)	35 (2.23%)	0.586	0.399	0.861	0.007
Primary sidedness	Left colon or rectum (N = 4827)	163 (3.38%)	1	ref.		
	Right colon (N = 1570)	35 (2.23%)	0.652	0.451	0.944	0.024
Primary localization (colon vs. rectum)	Colon (N = 4149)	107 (2.58%)	1	ref.		
	Rectum (N = 2964)	111 (3.74%)	1.47	1.122	1.925	0.005
Sample type	Surgery (N = 4452)	140 (3.14%)	1	ref.		
	Endoscopy (N = 1388)	45 (3.24%)	1.032	0.734	1.452	0.856
	Needle (N = 218)	12 (5.50%)	1.794	0.979	3.289	0.059
Sample origin (primary vs. metastatic)	Primary (N = 4968)	153 (3.08%)	1	ref.		
	Metastatic (N = 685)	30 (4.38%)	1.441	0.966	2.15	0.073
Sample origin (detailed)	Primary (N = 4968)	153 (3.08%)	1	ref.		
	Metastatic—other (N = 598)	23 (3.85%)	1.259	0.805	1.968	0.313
	Metastatic—local (N = 82)	5 (6.10%)	2.044	0.815	5.121	0.127
	Metastatic—CNS (N = 5)	2 (40.00%)	20.98	3.48	126.47	0.001
Biopsy vs. surgery for metastatic samples	Surgery (N = 854)	32 (3.75%)	1	ref.		
	Biopsy (N = 259)	14 (5.41%)	1.468	0.771	2.795	0.243

Right colon (proximal) included caecum, ascending colon, hepatic flexure, and transverse colon. Left colon (distal) included splenic flexure and descending and sigmoid colon. Rectum included rectosigmoid tumours. MANEC—mixed adeno-neuroendocrine cancer. *p*—for quantitative variables, Mann–Whitney test; for qualitative variables, chi-squared test or exact Fisher test. Expanded version available in the (Appendix A).

**Table 6 ijms-24-09073-t006:** Clinical and pathological factors that affected the rate of *KRAS* G12C mutation among other KRAS mutations.

Trait	Group	KRAS G12C	OR	95% CI		*p*
Sex	Male (N = 2200)	136 (6.18%)	1	ref.		
	Female (N = 1628)	101 (6.20%)	1.004	0.77	1.309	0.978
Age (years)	<40 (N = 65)	6 (9.23%)	1	ref.		
	41–50 (N = 203)	13 (6.40%)	0.673	0.245	1.848	0.442
	51–60 (N = 624)	36 (5.77%)	0.602	0.244	1.488	0.272
	61–70 (N = 1597)	111 (6.95%)	0.735	0.31	1.739	0.483
	71–80 (N = 1134)	64 (5.64%)	0.588	0.245	1.414	0.236
	Over 80 (N = 205)	7 (3.41%)	0.348	0.113	1.074	0.066
Histology	Partially mucinous (N = 405)	23 (5.68%)	1	ref.		
	Mucinous (N = 10)	0 (0.00%)	--	--	--	--
	Cribriform or comedo (N = 11)	2 (18.18%)	3.691	0.753	18.079	0.107
	MANEC (N = 2)	0 (0.00%)	--	--	--	--
	Other (N = 292)	16 (5.48%)	0.963	0.499	1.856	0.91
Primary tumour localization	Rectum (N = 1533)	111 (7.24%)	1	ref.		
	Left colon (N = 876)	51 (5.82%)	0.792	0.562	1.115	0.182
	Right colon (N = 819)	35 (4.27%)	0.572	0.387	0.845	0.005
Primary sidedness	Left colon or rectum (N = 2418)	163 (6.74%)	1	ref.		
	Right colon (N = 819)	35 (4.27%)	0.618	0.425	0.898	0.012
Primary localization (colon vs. rectum)	Colon (N = 2060)	107 (5.19%)	1	ref.		
	Rectum (N = 1533)	111 (7.24%)	1.425	1.083	1.874	0.011
Sample type	Surgery (N = 2292)	140 (6.11%)	1	ref.		
	Endoscopy (N = 699)	45 (6.44%)	1.058	0.748	1.497	0.752
	Needle (N = 102)	12 (11.76%)	2.05	1.096	3.833	0.025
Sample origin (primary vs. metastatic)	Primary (N = 2529)	153 (6.05%)	1	ref.		
	Metastatic (N = 341)	30 (8.80%)	1.498	0.995	2.255	0.053
Sample origin (detailed)	Primary (N = 2529)	153 (6.05%)	1	ref.		
	Metastatic—other (N = 293)	23 (7.85%)	1.323	0.838	2.087	0.229
	Metastatic—locoregional (N = 44)	5 (11.36%)	1.991	0.774	5.124	0.153
	Metastatic—CNS (N = 4)	2 (50.00%)	15.529	2.173	110.997	0.006
Biopsy vs. surgery for metastatic samples	Surgery (N = 450)	32 (7.11%)	1	ref.		
	Biopsy (N = 124)	14 (11.29%)	1.662	0.857	3.224	0.132

Right colon (proximal) included caecum, ascending colon, hepatic flexure, and transverse colon. Left colon (distal) included splenic flexure, descending and sigmoid colon. MANEC—mixed adeno-neuroendocrine cancer. *p*—univariate logistic regressions.

## Data Availability

The data presented in this study are available upon request from the corresponding author. The data are not publicly available due to ethical and legal concerns.

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
