# Peer review of "Clinical Characterization of Targetable Mutations (BRAF V600E and KRAS G12C) in Advanced Colorectal Cancer—A Nation-Wide Study"

_ijms, 2023, doi:10.3390/ijms24109073_

Round 1
Reviewer 1 Report
This is a very impressive study, which beautifully fits to the topic of the Special Issue “State-of-the-Art Molecular Oncology in Poland”. It describes two rare actionable mutations in a very large collection of colorectal carcinomas tested by a uniform methodology. Some important correlations between molecular and clinical features of the tumors were observed. This paper definitely deserves publication, although several issues need to be considered.
Please check the wording with the utmost level of accuracy and the use of the WORD spelling check (regular WORD software has the in-built function, which allows for elimination of typing errors). There are some avoidable mistakes (e.g., “patomechanisms” – line 43; “ccopathological” – line 250; “polish” instead of “Polish” – line 271).
To my knowledge, “Materials and Methods” section in the IJMS has to be located after the Discussion, at the end of the manuscript.
To my impression, the frequency of BRAF V600E in the studied population is not low. Western studies generally included somewhat older patients with colorectal cancer, which is probably related to higher life expectancy in Western Europe and North America (line 267). In addition to the factors, which are discussed in great detail by the authors, the proportion of patients aged above 80 years may be a contributing factor, given that BRAF mutations demonstrate gradual age-dependent frequency increase (for example, see Table 3 in this manuscript or published data sets). Only approximately 5% of the analyzed patients in this paper are aged above 80 years, which may affect the over frequency of BRAF lesions.
Please check the wording with the utmost level of accuracy and the use of the WORD spelling check (regular WORD software has the in-built function, which allows for elimination of typing errors). There are some avoidable mistakes (e.g., “patomechanisms” – line 43; “ccopathological” – line 250; “polish” instead of “Polish” – line 271).
Author Response
Thank you for the review and for all the comments regarding the manuscript’s shortcomings. We have addressed the outlined issues in a following manner:
Wording: The manuscript has been again reviewed in terms of spelling. Several errors were corrected including the ones commented on.
Materials and Methods: Thank you for bringing our attention to this requirement. The manuscript has been re-formatted accordingly.
BRAF V600E frequency: This is indeed factor that was not highlighted enough in the discussion. Changes have been included in lines 277-281 and 295-296 to better address this issue.

Reviewer 2 Report
The article : Clinical characterization of targetable mutations (BRAF V600E and KRAS G12C) in advanced colorectal cancer - a nation-wide study.”, submitted to the International Journal of Molecular Sciences, by Paweł M Potocki and colleagues report the findings about KRAS G12C and BRAF V600E mutations in Polish cohort. It is very impressive that the study contained more than 7000 patients records. However, there are some concerns from my side that should be addressed.
Major points:
1.
Many other KRAS mutations existing and have a role in CRC, for example: KRAS G12D, KRAS G12V, KRAS G13C. You mentioned in Table 1 and Table 4, other KRAS mutations. This raises the question: What percentage of these other mutations occur in this cohort?
I think the different KRAS mutations percentage should be represented as a pie chart diagram and a paragraph should be written about these mutation in the introduction part: around line 70. Highlight the mutations that will be discussed later, hopefully KRAS G12D and KRASG12C.
2.
According to other publications (doi:10.1038/s41417-022-00561-3., doi:10.1038/s41392-021-00780-4., doi:10.3390/cells10030667) KRAS G12D is the most common mutation in CRC.
This is a very complicated mutation, nowadays more and more research groups are working to find a way to inhibit it.
My suggestions are to conduct all of the analysis for the KRAS G12D mutation, because it could be very interesting (taken into consideration its difficulty to develop an inhibitor for this mutation) and add the KRAS G12C mutation datas to the figures/tables, then present the findings in the discussion.
3.
My recommendation (if it is possible) to do a Kaplan-Meier analysis for the KRAS G12C vs no KRAS G12C mutation cohort, KRAS G12D vs no KRAS G12D mutation cohort, and BRAF V600E vs no BRAF V600E mutation cohort and write about the findings in the discussion.
4. In my opinion you should add the M score too to the tables.
5. The data visualization is very superficial; the tables are too long. Here are some ideas to improve the figures and tables:
The Age datas from all the tables (Table 1, Table 3, Table 5, Table 6) maybe illustrated in a column diagram.
Sex datas might be represented in a pie chart (Table 1, Table 3, Table 5, Table 6).
Certain parts of the table should be in the supplementary files:
- Biopsy/surgery (Table 1, Table 3, Table 5, Table 6)
- Biopsy/surgery for primary tissues (Table 1, Table 2, Table 4)
- Biopsy/surgery for metastatic tissues (Table 1, Table 3, Table 5, Table 6)
- Grade (1-2 vs 3-4) from Table 1, Table 3, Table 5, Table 6
- N score (0 vs 1 vs 2) from Table 1, Table 2, Table 4
It would be very impressive if the region datas were plotted on a map with the corresponding data.
6. Figure 1 should be improved: the first figure can be converted into a pie chart, where different colors represent different cases.
7. After the proposed modifications and analyses, the summary and abstract should be rewritten and a new title may be considered.
Minor points:
The line 156 is in the middle of the page.
In the line 50 there is a typo: GRHB2 (should be corrected to GRB2)
Explanation of abbreviations is incomplete: OR, CI, Adenocarcinoma NOS
Description of the Figures, Tables should be clear and indicate every detail, the figure itself must be understandable with the description.
All significant data points/lines in tables and figures should be highlighted in color.
Please specify “group” in Table 3, Table 5, Table 6. Add a clear explanation about this in the table descriptions.
Sometimes the English text doesn't seem very professional, for exapmle line 301-302, 331-332.
Author Response
Many thanks for the review and for all the comments regarding the manuscript’s shortcomings. We have addressed the outlined issues in a following manner:
Major points:
- Thank you for the remark. Characterizing the population of patients with mutations not addressed in this study is very in line with our most immediate plans regarding this database. It is unfortunately not possible in this paper. As we are a small and unfunded team at the moment we have decided the data from a subset of our database (manageable within our current resources) to be published first as a proof of concept which will hopefully help us expand. Adding the data on the frequency of other KRAS/NRAS alterations would change and vastly expand the scope of this publication. There is a next analysis planned which is aimed at all of the mutations.
- As mentioned previously we are aware of the importance of other mutations in KRAS and plan to write a separate paper on this matter – hopefully this year.
- Thank you for a very good point. We do not have either PFS nor OS data available at the moment but we are working quite actively at obtaining those. We are aware that there are reports on prognostic differences between the patients with the most common RAS mutations and a scarcity of such in on the patients with rarer mutations. Therefore we see the need of investigating the impact of the mutation type on treatment outcomes in our cohort. This is for now outside of the scope of this study.
- Thank you for this remark. All the patients presented in this dataset were metastatic at the time of molecular profiling which is already mentioned in the materials & methods section. Unfortunately the clinical records available did not contain the data on timing of dissemination (synchronous vs metachronous) therefore we have not been able to include this factor in our analysis.
- Thank you for these suggestions. The tables have been restructured and shortened. Full versions were supplemented with a forest plot and moved to the supplementary data. The graphical presentation of the results has been reworked. Charts to depict the most significant correlations were added.
- Figure 1 has been reworked to include the pie chart
- As mentioned in previous answers: we respectfully highlight our inability to comply with all of the suggestions of the reviewer. Some of the suggestions (expanding the analysis to all mutations, survival data) are both beyond the scope of the current study and beyond our current means. We hope to be able to pursue more detailed analysis in the future works, therefore we have decided to leave the title and the abstract unchanged.
Minor points:
The line 156 has been reformatted
GRHB2 and several other typos have been corrected.
The abbreviations mentioned and several others have been explained.
The captions for figure and tables have been reworked
Significant datapoints in tables have been highlighted
Table groups and captions have been altered for clarity.
Reviewer 3 Report
This manuscript titled “Clinical characterization of targetable mutations (BRAF 2 V600E and KRAS G12C) in advanced colorectal cancer - a na-3 tion-wide study” by Potocki et al. The authors have performed a thorough clinical study on ~7600 patient samples to characterize the prevalence of BRAF V600E and KRAS mutations in colorectal cancers. The authors have done a detailed analysis and investigated the characteristics of CRC patients harboring these mutations. The key observations have highlighted the location, severity, gender specificity, vascular invasion, and metastasis of colorectal cancer. The Authors have identified a potential candidate population for BRAF mutation as the one with a high prevalence of BRAF V600E mutations with neuroendocrine components. Also, the other key finding of this study is the KRAS G12C mutation associated with the left-side intestine and brain metastasis of CRC. This study further provides evidence for using KRAS and BRAF target therapies as a potential future for CRC patients. The editors should consider this manuscript for publication since it is a critical study highlighting key oncogenic mutations and their clinical relevance in diagnosis and treatment in a relatively large patient cohort. There are a few edits that can make the impact of this manuscript to be eligible for publication in IJMS.
1. The authors have provided most data in tabulated forms and have provided critical differential analysis in the texts. The essential data should be presented as charts or graphs, which would be easier to understand and interpret the data and conclusions.
2. The authors have missed references in the introduction and result sections, e.g., lines 38, 97, 325, 327, etc. Authors should mention appropriate references, which they already mentioned in the reference section of the text.
3. A few typos in the manuscript should be addressed. e.g., lines 41, 45, 60, etc.
4. The authors have provided information about present developed therapies for BRAF, EGFR, and MEK in CRC and mCRC. It should be interesting if the authors could explain their findings and use of presently available target therapies for CRC patients. It is essential to comment on how this study could be helpful for Oncology clinicians to advocate for suitable therapies for their CRC patients based on the KRAS and BRAF mutation status of their cancers.
The quality of English in the article is primarily fine, only in a few places. It is challenging to understand due to some misspelled words; otherwise, it is good.
Author Response
Many thanks for the review and for all the comments regarding the manuscript’s shortcomings. We have addressed the outlined issues in a following manner:
- The graphical presentation of the results has been reworked. Charts to depict the most significant correlations were added. Tables were shortened. Full versions were supplemented with a forest plot and moved to the supplementary data.
- The missed references have been added.
- The manuscript has been again reviewed in terms of spelling. Several errors were corrected including the ones commented on.
- The information on most recent FDA tissue agnostic approval for BRAF-mutated cancer was added in the discussion (line 303 after corrections)
Round 2
Reviewer 2 Report
Thank you for your answer, I understand your point of view.
However, some minor points should be addressed.
In the first figure, the number of cases should be put in parentheses for example: (102), because it can be misunderstood due to the comma.
Figure 4 (Geographical distributions) should be in the manuscript not in the supplementary files.
Author Response
Thank you again for the review and for your understanding for our choice of methodology and publication strategy. We have addressed the outlined issues in a following manner:
The captions in the first figure were formatted to more clearly distinguish between absolute numbers and percent values. I took liberty to use parentheses for percent values and not the absolute numbers as this is the convention used throughout the paper.
The figure regarding the geographical distribution has been moved from supplementary data to the main text.